

# Automatic differentiation applied to excitations with projected entangled pair states

Boris Ponsioen[1*], Fakher F. Assaad[2] and Philippe Corboz[1]

**1** Institute for Theoretical Physics Amsterdam and Delta Institute for Theoretical Physics, University of Amsterdam, Science Park 904, 1098 XH Amsterdam, The Netherlands
**2** Institut für Theoretische Physik und Astrophysik and Würzburg-Dresden Cluster of Excellence ct.qmat, Universität Würzburg, Am Hubland, D-97074 Würzburg, Germany

⋆ b.g.t.ponsioen@uva.nl

## Abstract

The excitation ansatz for tensor networks is a powerful tool for simulating the low-lying quasiparticle excitations above ground states of strongly correlated quantum many-body systems. Recently, the two-dimensional tensor network class of infinite projected entangled-pair states gained new ground state optimization methods based on automatic differentiation, which are at the same time highly accurate and simple to implement. Naturally, the question arises whether these new ideas can also be used to optimize the excitation ansatz, which has recently been implemented in two dimensions as well. In this paper, we describe a straightforward way to reimplement the framework for excitations using automatic differentiation, and demonstrate its performance for the Hubbard model at half filling.



# 1 Introduction

As both a computational tool and a conceptual framework for simulating and understanding the physics of strongly correlated quantum systems, tensor networks have gained a prominent role in recent years. In two dimensions, the infinite projected entangled-pair state (iPEPS) ansatz [1–6] is the natural extension of the one dimensional matrix product states (MPS) [7–9] that have been widely used in many applications. While in early years the optimization of iPEPS for ground states and the evaluation of expectation values was considered too costly in comparison to efficient quasi-two-dimensional MPS-based approaches, both the development of efficient and powerful algorithms as well as the improvement in available computational power have made accurate applications possible. These methods, which are able to run on now commonly available hardware, generally require significant developmental investment and have therefore not yet been adopted by a large community.

One recent advancement that promises a lower barrier for ground-state optimization of iPEPS is the application of automatic differentiation (AD) [10], a technique that is well known in the context of machine learning. The main advantage of this approach is that, if an algorithm is built from computation steps for which the analytical derivative can be explicitly implemented, only the code for evaluating the cost function is required, in our case the energy expectation value. By applying the chain rule of differentiation, the associated computational graph can be traversed in an automized manner, such that the gradient of the energy with respect to the variational parameters can be computed without any further programming effort. The application of AD in iPEPS ground-state simulations, recently also in the highly nontrivial study of the phase transition in the $J_1 - J_2$ model from the ordered to a quantum spin liquid phase [11], offers a promising new direction.

While the iPEPS ansatz is best known for its use in ground-state simulations, also recently new extensions have emerged that target low-lying excited states [12] as well as thermal states [13–19]. The excitation ansatz for iPEPS, based on its one-dimensional MPS equivalent [20–23], offers the possibility to accurately simulate quasiparticle excitations on top of a strongly correlated ground state. After its original development [12] it has been extended to more general spin models [24] and fermionic models [25], the latter using a new implementation based on the corner transfer matrix (CTM) [26–29] contraction method.

Here we will show how the infinite summations required to contract the excited-state iPEPS network can be performed in a different way, leading to a simpler implementation and lower computational cost. Then we detail how to use the concepts of AD in order to find to optimized lowest-lying excited states, using a fully U(1)-symmetric differentiable tensor contraction implementation. Additionally, we propose a variant based on the fixed-point of the CTM algorithm, introduced earlier for ground-state simulations [10], leading to a large further reduction in memory cost.

We demonstrate the performance of the new algorithm on a nontrivial example of the

calculation of the charge gap in the two-dimensional Hubbard model and compare to results from Quantum Monte Carlo (QMC). We also present results and a comparison for the spectral function at intermediate interaction strengths.

The first section of this paper introduces the iPEPS concepts that form the foundation of the algorithm. Then we continue with the formulation of the iPEPS excitation ansatz and a detailed description of the CTM summation scheme. In Secs. 2.3 and 2.4 we introduce the application of AD in the context of excitations and provide pseudocode outlines of the general algorithm as well as the fixed-point variant. Finally we present the results obtained for the 2D Hubbard model at half filling with both iPEPS and QMC, with details regarding the QMC computations in Appendix A. In Appendix B we describe additional technical details relevant for implementing our algorithm. For the iPEPS algorithm we additionally provide a fully functional basic version of the code [30], including a ground-state optimization method.

## 2 Methods

### 2.1 iPEPS

Projected entangled-pair states (PEPS) [1–6], sometimes also known as tensor product states, form the natural extension of matrix product states to two dimensions. In the infinite version (iPEPS) [31], the ansatz consists of an infinite network of order-5 tensors, each containing $dD^4$ variational parameters. The different dimensions $d$ and $D$ refer to the local Hilbert space on a single site and the bond dimension, respectively, the latter of which controls the accuracy of the ansatz. For simplicity, in this paper we limit the discussion to translationally invariant systems, with a one-site unit cell with a single tensor $A$ that is repeated on the lattice, though the implementation can be easily extended to arbitrary unit cell sizes [25].

Since the excitation ansatz is constructed from a ground state, first the optimized ground-state $A$ tensor has to be obtained. Recently, it was shown [10] that AD can be employed to perform accurate and efficient ground-state simulations. We have used this idea here in combination with symmetric tensors [32, 33], where we exploit the global Abelian symmetry of the Hubbard model in order to greatly speed up the computations.

### 2.2 iPEPS excitation ansatz

The iPEPS excitation ansatz, first described in [12] and extended in [24, 25], can be used to approximate quasiparticles localized in momentum space. Starting from the ground state, we modify the iPEPS by changing one of the ground-state $A$ tensors by a different tensor $B$ on a location $\boldsymbol{x} = (i, j)$:

$$|\Psi_0(A)\rangle \ \rightarrow \ |\Phi(A, B)_{\boldsymbol{x}}\rangle, \tag{1}$$

and pictured in Fig. 1. Consequently, the excited state with momentum $\boldsymbol{k} = (k_x, k_y)$ is constructed by performing a summation with appropriate phase factors

$$|\Phi(B)_k\rangle = \sum_{\boldsymbol{x}} e^{i\boldsymbol{k}\cdot\boldsymbol{x}} |\Phi(B)_{\boldsymbol{x}}\rangle. \tag{2}$$

The goal is now to find the optimal variational parameters in tensor $B$ with the $A$ tensors kept fixed. Minimizing the energy under the constraint that the wavefunction is normalized writes

$$\frac{\partial}{\partial B^\dagger} \left[ \langle \Phi(B)_k | \mathcal{H} | \Phi(B)_k \rangle - \omega_k (\langle \Phi(B)_k | \Phi(B)_k \rangle - 1) \right] = 0. \tag{3}$$

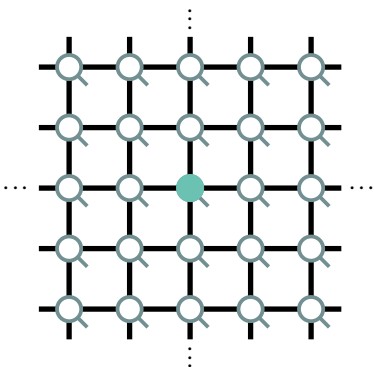

Figure 1: Schematic picture of an excitation tensor in the center of an infinite PEPS.

The first term corresponds to the triple infinite sum

$$\frac{\partial}{\partial B^{\dagger}} \left\langle \Phi(B^{\dagger})_k \middle| \mathcal{H} \middle| \Phi(B)_k \right\rangle = \frac{\partial}{\partial B^{\dagger}} \sum_{x_1, x_2, j} e^{ik\cdot(-x_1+x_2)} \left\langle \Phi(B^{\dagger})_{x_1} \middle| h_j \middle| \Phi(B)_{x_2} \right\rangle, \qquad (4)$$

where $\mathcal{H} \equiv \sum_j h_j$ refers to a summation of local Hamiltonian terms on different bonds $j$. We can rewrite the expectation value $\left\langle \Phi(B^{\dagger})_k \middle| \mathcal{H} \middle| \Phi(B)_k \right\rangle$ as $\vec{\mathbf{B}}^{\dagger} \mathbb{H}_k \vec{\mathbf{B}}$, defining the effective Hamiltonian matrix, $\mathbb{H}_k$, which corresponds to the tensor network representing this expectation value with the $B$ and $B^{\dagger}$ tensors removed and reshaped into a matrix, and with $\vec{\mathbf{B}}$ and $\vec{\mathbf{B}}^{\dagger}$ being the tensors $B$ and $B^{\dagger}$ reshaped into vectors, respectively. In a similar fashion, we can define an effective norm matrix, $\mathbb{N}_k$, with $\left\langle \Phi(B^{\dagger})_k | \Phi(B)_k \right\rangle = \vec{\mathbf{B}}^{\dagger} \mathbb{N}_k \vec{\mathbf{B}}$, involving a double-infinite sum. Solving Eq. 3 boils down to solving the generalized eigenvalue problem for $\vec{\mathbf{B}}$,[1]

$$\mathbb{H}_k \vec{\mathbf{B}} = \omega_k \mathbb{N}_k \vec{\mathbf{B}}. \qquad (5)$$

The main technical challenge is to evaluate the triple infinite sum in Eq. 4. Taking the derivative with respect to the $B^{\dagger}$ tensor removes the tensor from the network, leaving a "hole" in its place. Thus, after taking the derivative we have an infinite sum over the location of the $B$ tensor, the Hamiltonian terms, and the location of the hole. By exploiting translational invariance we can eliminate one of these sums. In the previous implementations in Refs. [24, 25] the sum is taken over the former two, with the location of the hole fixed at the center. Here we adopt a different strategy, namely that we eliminate the sum over Hamiltonian terms instead, which has the advantage that the summation of 2-body terms (which is technically more challenging and computationally more expensive than the summation over the B tensors) can be avoided. The question is then how the summation over the hole can be performed. While this summation could be implemented in a manual fashion as done in Ref. [34] for ground state calculations, here we will show that a much simpler approach is offered by automatic differentiation (AD). We simply perform a summation over the $B$ and $B^{\dagger}$ tensors and compute the derivative with respect to $B^{\dagger}$ in an automatized fashion using AD. This results in a code which is substantially simpler, faster, and much more easy to generalize to Hamiltonians beyond nearest-neighbor interactions.

## 2.3 Automatic differentiation

Although its name might suggest an opaque and automized computational tool, AD is nothing more than a programmatic way to apply the chain rule of differentiation. Most commonly used

---

[1]In practice, we need to restrict the $B$ tensor to a subspace in which the modes with small norm have been removed. We discuss the dependence on the subspace size in Appendix B.

in the field of machine learning, where it forms the backbone of most optimization methods, AD requires the explicit implementation of the derivatives of basic operations [2]. If all steps in a given algorithm can be reduced to combinations of these basic operations, the AD code is able to construct the gradient of the output of the full algorithm with respect to its inputs. Evidently, if a particular algorithm is made of operations of which the derivative is already known, the implementation effort required of the programmer is extraordinarily low.

Recently AD techniques have been applied to the optimization problem of iPEPS [10]. Considerable effort has been put into development of powerful yet efficient optimization algorithms for iPEPS in recent years, from very inexpensive [35] though generally less accurate or more balanced [1,3,31] imaginary time evolution methods to powerful though more expensive variational optimization approaches [36,37]. While these methods form a broad and widely applicable set of tools, AD offers a compelling alternative since it essentially does not require the implementation of any optimization algorithm at all, but only a routine for contracting the network and computing expectation values.

AD iPEPS ground-state optimization with a restricted U(1) symmetric ansatz has been recently used [11], though this restriction only applied to the variational parameters and not to the full computation. We have implemented a code that makes efficient use of the reduced variational subspace for all tensor contractions, which is commonly used in many other tensor network studies [32, 33], including also the excitation ansatz [23, 25], while retaining the gradient tracking capability of the algorithm.

## 2.4 iPEPS excitations with AD

### 2.4.1 Three point functions with CTM

The three point functions in Eq. 4, where the Hamiltonian is chosen to be fixed in the center, can be evaluated using CTM. First we will show how to perform the double sums over the $B$ tensors in the bra and ket layers, which is a simplified variant of the technique introduced in [25].

The CTM algorithm aims to find the set of boundary tensors $\{C_i, T_i\}$ ($i = 1, 2, 3, 4$) that best approximates the infinite environment of a double-layer iPEPS (i.e. the norm of the wavefunction)

$$\langle \Psi | \Psi \rangle = \quad \cdots \quad \approx \tag{6}$$

where each node in the left figure represents a pair of ground state $\{A, A^\dagger\}$ tensors:

$$\tag{7}$$

---

[2]*Reverse-mode* AD, also known as backpropagation, requires the implementation of the application of a vector on the derivative of each operation $\mathbf{v} \cdot \frac{\partial f(\mathbf{x})}{\partial \mathbf{x}}$, known as a vector-Jacobian product. This type of differentiation is more efficient in our case than *forward-mode* AD, which uses Jacobian-vector products instead, since the input space is much larger than the output (a scalar).

The accuracy of the CTM algorithm is systematically controlled by the boundary bond dimension $\chi$ of the boundary tensors (thick black lines).

If we now define double-layer versions of excitation $B$ and $B^\dagger$ tensors by light and dark blue colors, respectively,

$$\tag{8}$$

and their combined version

$$\tag{9}$$

the goal of the algorithm is to compute additional boundary tensors that contain sums of these excitation tensors.

For example, the left row transfer matrix $BT_4$ contains all terms where a $B$ tensor is positioned left of the center site:

$$\tag{10}$$

Similarly, we define the other transfer matrices $\{BC_i, BT_i\}$ that contain $B$ tensors, as well as those with $B^\dagger$ tensors.

Lastly, there are boundary tensors that correspond to terms in which there are both a $B$ and a $B^\dagger$ tensor present in the *same* region, denoted by $\{BB^\dagger C_i, BB^\dagger T_i\}$. Such tensors can be computed by combining the other boundary tensors, as shown in Sec. 2.4.2.

We can then evaluate the energy of the excited state, given by the three point function in Eq. 4, by summing over all possible combinations of these boundary tensors, and placing a Hamiltonian term on the central bonds.

The diagrams corresponding to a horizontal local Hamiltonian term $h_{hor}$ are the following:

$$\sum_{x_1,x_2} e^{i\boldsymbol{k}\cdot(-\boldsymbol{x}_1+\boldsymbol{x}_2)} \left\langle \Phi(B^\dagger)_{x_1} \middle| h_{hor} \middle| \Phi(B)_{x_2} \right\rangle =$$

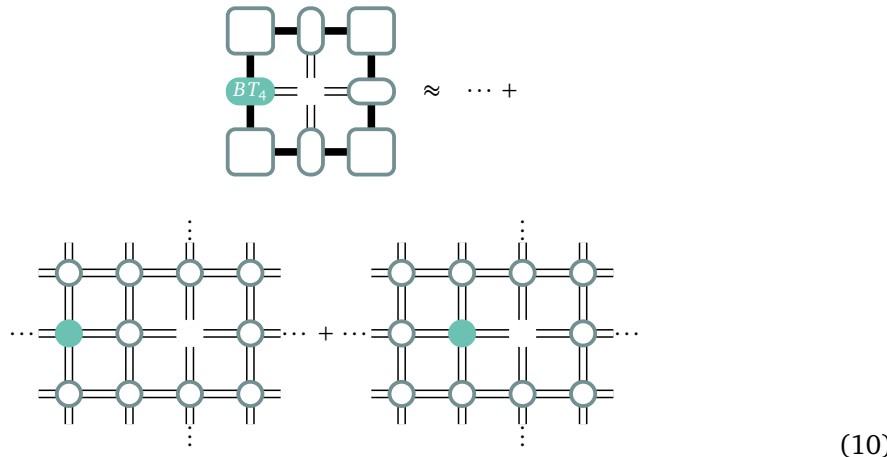

$$\tag{11}$$

A similar evaluation has to be performed for the vertical Hamiltonian term. (In case of a larger unit cell, a separate evaluation for each bond in the unit cell has to be performed).

### 2.4.2 CTM summation

In this section we will describe how to obtain the optimized boundary tensors that contain infinite summations over $B$ and $B^\dagger$ tensors. The standard CTM procedure for computing boundary tensors for the *norm* environment, meaning no $B$ or $B^\dagger$ tensors present, is done by iteratively absorbing a new row or column of sites into the boundary tensors until convergence.

For the left row transfer matrix $T_4$, the update step referred to as the *left move,* can be diagrammatically described as follows:

$$\text{(12)}$$

The $T_4'$ tensor on the left-hand side is the updated boundary tensor, while on the right a site (i.e. a pair of ground state tensors) is contracted with the boundary tensor $T_4$ of the previous iteration, and *projectors* (black triangles) are applied in order to truncate the result to the most relevant subspace [29, 38, 39]. These projectors reduce the bond dimension of the boundary tensor, which grows to $\chi D^2$ by absorbing the new site, back down to $\chi$. For the summation CTM we use the same projectors as for the regular CTM, for which we follow the procedure of Ref. [29], with the additional remark that the QR decomposition used in Ref. [29] is not necessary for the computations [40].

The top-left corner transfer matrix $C_1$ is updated during the left move by absorbing a half-column of sites, contained in the column transfer matrix $T_1$:

$$\text{(13)}$$

Updating the $BT_4$ row transfer matrix, containing terms with a $B$ tensor located in the left half row, is done by combining the absorption of a regular site in the previous $BT_4$ tensor with the absorption of a $B$ tensor into the $T_4$ boundary tensor:

$$\text{(14)}$$

Note that the appropriate phase factors have to be taken into account.

The update step for the boundary tensors that contain sums over $B^\dagger$ tensors is equivalent:

$$\text{(15)}$$

Finally, the update of boundary tensors that contain combinations of both a $B$ and a $B^\dagger$ tensor is an extension of this idea, where now more combinations have to be taken into account:

$$ (16) $$

The update steps for the corner tensors are analogous; the left move for the $BC_1$ tensor is as follows:

$$ (17) $$

### 2.4.3 Computing the lowest excited states

The procedure in the previous sections, which computes the expectation value $\tilde{E} = \vec{\mathbf{B}}^\dagger \mathbb{H}_k \vec{\mathbf{B}}$, can be viewed as a computational graph. We denote the full function by $f$, such that the unnormalized energy is given by $\tilde{E} = f(\vec{\mathbf{B}}^\dagger, \vec{\mathbf{B}})$, with intermediate steps $T_i$:

$$ f : (\vec{\mathbf{B}}^\dagger, \vec{\mathbf{B}}) \to T_1 \to T_2 \to \cdots \to T_n \to \tilde{E} . \qquad (18) $$

Importantly, for each step in the algorithm $T_i$ the associated derivative with respect to the output of the previous step $\frac{\partial T_i}{\partial T_{i-1}}$ is explicitly known [3]. The automatic differentiation framework tracks all steps in the forward pass of $f$, building up the computational graph, and then starts from the output to apply the chain rule in order to trace back through the graph and arrive at the full gradient:

$$ \mathbb{H}_k \vec{\mathbf{B}} = \frac{\partial f}{\partial \vec{\mathbf{B}}^\dagger} = \frac{\partial f}{\partial T_i} \frac{\partial T_i}{\partial T_{i-1}} \cdots \frac{\partial T_2}{\partial T_1} \frac{\partial T_1}{\partial \vec{\mathbf{B}}^\dagger} = \frac{\partial}{\partial B^\dagger} \sum_{x_1, x_2, j} e^{ik\cdot(-x_1 + x_2)} \left\langle \Phi(B^\dagger)_{x_1} \middle| h_j \middle| \Phi(B)_{x_2} \right\rangle $$
$$ \propto \frac{\partial}{\partial B^\dagger} \sum_{\substack{x_1, x_2 \\ j \in \text{unit cell}}} e^{ik\cdot(-x_1 + x_2)} \left\langle \Phi(B^\dagger)_{x_1} \middle| h_j \middle| \Phi(B)_{x_2} \right\rangle , \quad (19) $$

where we use translational invariance to reduce the infinite summation over the Hamiltonian terms to a sum over non-equivalent terms on bonds in the unit cell, as described in Sec. 2.2.

The action of the effective norm matrix $\mathbb{N}_k \vec{\mathbf{B}}$, which is computed in the same way as in our previous implementation [25], requires only the environment tensors $\{BC_i, BT_i\}$, which are created during the forward pass of $f$.

Once the actions of the effective matrices are implemented, an iterative solver for generalized eigenvalue problems can be used to compute the lowest excited state for a fixed value of k, described in Algorithm 1. Higher-energy excitations can be obtained by solving the generalized eigenvalue problem for the $k$ lowest eigenvalues $w_\alpha$, resulting in different tensors $B_\alpha$, where $\alpha = \{1, 2, \ldots, k\}$ labels the eigenstates.

---

[3]The basic steps performed in the CTM summation algorithm are the same as in the regular CTM algorithm for ground states, and therefore we refer to [10] for the implementation of the derivatives.

---

**Algorithm 1** Compute lowest excited state

---

```
Require:
    A, A†                                              ▷ ground-state tensors

 1: function apply_N_eff(B, B†)
        ▷ C contains the boundary tensors
 2:     C ← summation_ctm(B, B†)                        ▷ Equations 12 to 17
 3:     return compute_norm_grad(B, B†, C)                  ▷ Equation 10
 4: end function

 5: function apply_H_eff(B, B†)
        ▷ The boundary tensors C can be reused from apply_N_eff
 6:     C ← summation_ctm(B, B†)
 7:     E ← compute_energy(B, B†, C)                        ▷ Equation 11
        ▷ Compute the gradient of E with respect to B† via AD
 8:     return run_backward(E, var=B†)
 9: end function

        ▷ Solve the generalized eigenvalue problem with an iterative solver
10: ω ← generalized_eigensolver(apply_H_eff, apply_N_eff)
```

---

Although the AD scheme of Algorithm 1 avoids the larger computational cost of previous iPEPS excitation algorithms, it requires intermediate results in the forward pass to be stored. This leads to a generally higher memory cost, which scales linearly with the number of CTM iterations. One way to reduce the cost is by using checkpoints, which turns off the storage of selected operations and recomputes their results in the backward pass, effectively trading memory cost for computation time. In the next section, we introduce an alternative formulation of the algorithm, for which the memory cost is now proportional to only a single CTM step, regardless of how many are required.

### 2.4.4 Fixed-point variant

In the first implementation of automatic differentiation in iPEPS [10], a variant of the standard CTM implementation for ground states was proposed, based on the fixed point properties of the algorithm. Instead of performing a number of forward CTM iterations with gradient tracking enabled and evaluation of the energy with a subsequent backward traversal through the computational graph to obtain the gradient, this variant only requires gradient tracking from a single CTM step: as the CTM algorithm converges, the function that performs a single CTM step from a set of boundary tensors and site tensors approaches a fixed point, after which the boundary tensors remain invariant. At this fixed point, the implicit function theorem [41] can be utilized to compute the gradient. Essentially, by recycling the fixed-point boundary tensors and projectors at the fixed point, an arbitrary number of backward steps can be performed until the gradient reaches convergence. This alternative provides a significant improvement over the standard scheme. Since only the tensors and projectors from the last CTM iterations need to be stored, it reduces the memory cost by a factor $N_{CTM}$, corresponding to the number of CTM steps required to converge the gradient, compared to the standard scheme.

In [10] the authors mention potential issues with the application of the fixed-point scheme to iPEPS ground-state simulations due to the gauge freedom inherent to the CTM RG transformations. We found that when a consistent fixed convention for the signs in the SVD oper-

---

**Algorithm 2** Fixed-point algorithm

---

```
Require:
    A,A†                                          ▷ ground-state tensors
    apply_N_eff                                   ▷ same as in Algorithm 1

    Definitions
 1: 𝒮:  summation_ctm_step                        ▷ single step of the summation CTM
 2: vjp: vector-Jacobian product                  ▷ obtained through backward AD

 3: function converge_boundaries(B, B†)
 4:     with (disable gradient tracking)
 5:         𝒞 ← initialize_boundaries(B, B†)
 6:         while 𝒞 not converged do
 7:             𝒞 ← 𝒮(B, B†, 𝒞)
 8:         end while
 9:     end with
10:     return 𝒞
11: end function

12: function apply_H_eff_fixed_point(B, B†, 𝒞)
13:     with (enable gradient tracking)              ▷ single tracked step
14:         𝒞 ← 𝒮(B, B†, 𝒞)
15:         E ← compute_energy(B, B†, 𝒞)             ▷ Equation 11
16:     end with

17:     g ← vjp(compute_energy,var=𝒞,vector=1)                        ▷ ∂E/∂𝒞
18:     result ← 0
19:     while result not converged do                ▷ geometric sum
20:         g ← vjp(𝒮,var=𝒞,vector=g)          ▷ ∂E/∂𝒞 [∂𝒮(𝒞,B,B†)/∂𝒞]ⁿ
21:         result ← result + vjp(𝒮,var=B†,vector=g)   ▷ g ∂𝒮(𝒞,B,B†)/∂B†
22:     end while
23:     return result
24: end function

        ▷ Obtain all converged boundary tensors without gradient tracking
25: 𝒞 ← converge_boundaries(B, B†)

        ▷ Solve the generalized eigenvalue problem with an iterative solver
26: ω ← generalized_eigensolver(apply_H_eff_fixed_point, apply_N_eff)
```

- Line 17: $\frac{\partial E}{\partial \mathcal{C}}$
- Line 19: geometric sum
- Line 20: $\frac{\partial E}{\partial \mathcal{C}}\left[\frac{\partial \mathcal{S}(\mathcal{C},B,B^\dagger)}{\partial \mathcal{C}}\right]^n$
- Line 21: $g\,\frac{\partial \mathcal{S}(\mathcal{C},B,B^\dagger)}{\partial B^\dagger}$

---

ations [4] is taken, the CTM projectors and boundary tensors converge element-wise. In each simulation, we take special care to compare the CTM projectors element-wise to verify the convergence.

The fixed-point version of the excitations algorithm is outlined in Algorithm 2, and can be

---

[4]In a singular value decomposition $M = UsV^\dagger$, the overall sign of each singular vector is arbitrary, since any sign change in a row of $U$ can be compensated by a sign change in a column of $V^\dagger$. We have chosen a convention in which the sign of the largest element of each singular vector in $U$ is positive. Special care should be taken if the singular value spectrum becomes degenerate, however, in our simulations this never turned out to be the case.

summarized as

$$\frac{\partial \tilde{E}}{\partial B^{\dagger}} = \sum_n \frac{\partial \tilde{E}}{\partial \mathcal{C}} \left[ \frac{\partial \mathcal{S}(\mathcal{C}, B, B^{\dagger})}{\partial \mathcal{C}} \right]^n \frac{\partial \mathcal{S}(\mathcal{C}, B, B^{\dagger})}{\partial B^{\dagger}}, \tag{20}$$

with $\mathcal{S}$ representing a single CTM iteration, taking a set of boundary tensors $\mathcal{C}$ and site tensors $B$ as input and returning a new set of boundary tensors.

## 3 Results

### 3.1 Charge gap in the half-filled Hubbard model

As a challenging benchmark example we consider the 2D Hubbard model on a square lattice given by the Hamiltonian

$$\hat{H} = -t \sum_{<i,j>,\sigma} \hat{c}^{\dagger}_{i,\sigma} \hat{c}_{j,\sigma} + \text{h.c.} + U \sum_i \hat{n}_{i,\uparrow} \hat{n}_{i,\downarrow}, \tag{21}$$

where $t$ is the nearest-neighbor hopping amplitude and $U > 0$ the on-site Coulomb repulsion, $\hat{c}^{\dagger}_{i,\sigma}$ and $\hat{c}_{i,\sigma}$ are the creation and annihilation operators for a fermion of spin $\sigma$ on site $i$, respectively, and $\hat{n}_{i,\sigma} \equiv \hat{c}^{\dagger}_{i,\sigma} \hat{c}_{i,\sigma}$ is the density operator. The 2D Hubbard model has been a subject of intense theoretical a numerical studies over the last decades, in particular due to its relevance to high-$T_c$ superconductivity, see Refs. [42, 43] for a recent review. Here we focus on the case of half-filling with $\langle \hat{n}_i \rangle = \langle \hat{n}_{i,\uparrow} + \hat{n}_{i,\downarrow} \rangle = 1$, where the ground state has a charge gap for $U > 0$ [44–48] and exhibits antiferromagnetic long-range order [49, 50].

The study of the low-lying single-hole excitations forms an important test case for our method. The removal of one particle in an antiferromagnetic background leads to collective excitations in the form of spinon-chargon combinations or polarons [51–57] which are directly experimentally observable in angle-resolved photoemission spectroscopy (ARPES). The absence of a negative-sign problem at half-filling allows for unbiased numerically exact results with Quantum Monte Carlo (QMC) techniques, which we compare our results to. In Appendix A we include details on how the QMC results are obtained. Additionally we make comparison with the recent state-of-the-art result from Ref. [58] for $U/t = 4$.

In order to obtain the iPEPS results, here for bond dimensions $D = 4, 5, 6$, accurate ground-state tensors have to be obtained first, for which we use a conjugate gradient optimizer with the gradient computed using automatic differentiation. Since the ground state has a partially broken translational symmetry due to antiferromagnetic order, we use a $2 \times 2$ unit cell with two different tensors arranged in a checkerboard pattern. From the optimized ground-state tensors, we use the excitation algorithm to obtain the optimized $B$ tensors and lowest eigenvalues. In order for the generalized eigenvalue problem of Eq. 5 to be well-defined, we work in a reduced basis for the $B$ tensors, as described in Appendix B. For the CTM algorithm we checked that the results are converged in the boundary bond dimension $\chi$. [5]

Our results for the gap of the hole (or charge) excitation as a function of $U/t$ are shown in Fig. 2. We observe a close correspondence between iPEPS and the QMC results for a wide range of $U/t$. For $U/t \leq 8$ the iPEPS data approaches the extrapolated QMC data in a systematic way with increasing bond dimension. For larger values of $U/t$ obtaining a reliable result with QMC turns out to be an exponentially hard problem, despite the fact that there is no sign problem. The reason is that the tail of the single particle Green function $G(\tau)$, from which

---

[5]In practice, we check multiple values of $\chi$ and make sure that the finite-$\chi$ effect on the results is much smaller than the error due to the finite bond dimension $D$. For the simulations in this paper, we found that for simulations with $D \leq 5$ a value of $\chi \equiv 80$ was sufficient, while for $D = 6$ we pushed to $\chi = 200$ ($U/t = 4$) and $\chi = 160$ ($U/t = 12$) in order to get stable results.

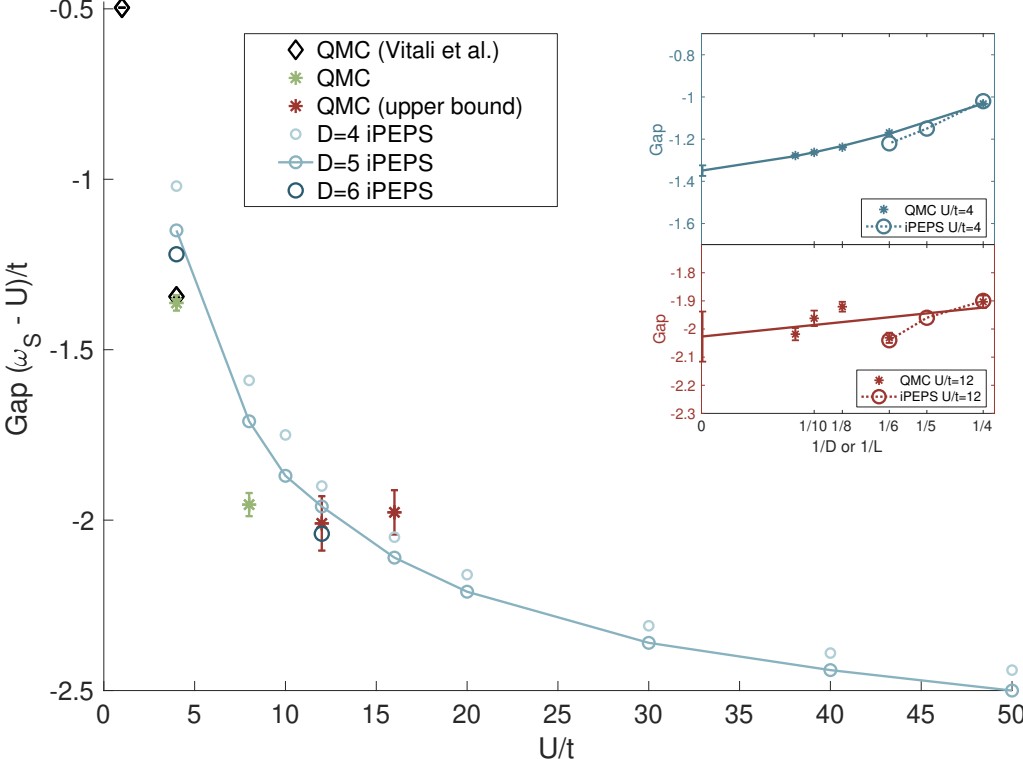

Figure 2: Comparison of iPEPS results for the gap at various values of $U/t$ with our extrapolated QMC results and those of Vitali et al. [58]. On the y axis, $\omega_S$ refers to the lowest eigenvalue at the $S = (\pi/2, \pi/2)$ point, and we applied a shift of $-U/t$ to better visualize the energy differences over the large range of $U/t$ on the x axis. For the green QMC points the data is well converged, while the results shown in red are an upper bound of the true gap due to an exponential scaling of the computational cost (see Appendix A for more details). Inset: Comparison of finite bond-dimension iPEPS and finite size QMC data for the gap at $U/t = 4$ (blue data) and $U/t = 12$ (red data).

the gap is extracted at large $\tau$, is exponentially suppressed with $U$, such that the number of Monte Carlo samples needed to resolve the tail grows exponentially with $U/t$, see Appendix A for details. This is also reflected in the irregular finite size effects in the QMC result for the gap shown in the inset of Fig. 2 for $U/t = 12$ (whereas the gap for $U/t = 4$ is monotonously decreasing with increasing system size). As a consequence QMC can only provide an upper bound for the value of the gap for large $U/t$ (shown in red in Fig. 2). The iPEPS simulations, in contrast, do not suffer from this limitation, and the gap can be extracted in a systematic way even at very large values of $U/t$. In fact, the dependence on $D$ is found to become weaker with increasing $U/t$. This is a natural consequence of the fact that the ground state becomes increasingly more entangled when approaching the noninteracting $U/t = 0$ limit, where the entanglement entropy exhibits a logarithmic correction to the area law [59,60], and hence the iPEPS calculations become increasingly challenging as we approach that limit. We note that, while the data systematically improves with increasing $D$, we have not attempted to extrapolate the gap to the infinite $D$ limit based on only 2-3 data points, also because the functional form of how the gap depends on $D$ is not known a priori.

## 3.2 Spectral function

The dispersion of the low-lying excitations can be computed for any point in the Brillouin zone using our method. In addition, we can study the hole spectral function given by

$$
\begin{aligned}
A_{hl}(\omega, \boldsymbol{k}) &= \sum_{\sigma} \int \mathrm{d}t \, e^{-i\omega t} \langle \Psi_0 | \hat{c}^{\dagger}_{\sigma,\boldsymbol{k}}(0) \hat{c}_{\sigma,\boldsymbol{k}}(t) | \Psi_0 \rangle \\
&= \sum_{\sigma} \int \mathrm{d}t \, e^{-i\omega t} \langle \Psi_0 | \hat{c}^{\dagger}_{\sigma,\boldsymbol{k}}(0) e^{i\hat{\mathcal{H}}t} \hat{c}_{\sigma,\boldsymbol{k}}(0) e^{-i\hat{\mathcal{H}}t} | \Psi_0 \rangle \\
&= \sum_{\sigma} \int \mathrm{d}t \, e^{-i\omega t} e^{-iE_0 t} \langle \Psi_0 | \hat{c}^{\dagger}_{\sigma,\boldsymbol{k}}(0) e^{i\hat{\mathcal{H}}t} \hat{c}_{\sigma,\boldsymbol{k}}(0) | \Psi_0 \rangle \,.
\end{aligned}
\tag{22}
$$

By inserting a complete basis of single-particle states $\sum_{\alpha} |\alpha\rangle\langle\alpha|$ with momentum $\boldsymbol{k}$ we obtain

$$
\begin{aligned}
A_{hl}(\omega, \boldsymbol{k}) &= \sum_{\sigma} \sum_{\alpha} \int \mathrm{d}t \, e^{-i\omega t} e^{i(E_\alpha - E_0)t} \langle \Psi_0 | \hat{c}^{\dagger}_{\sigma,\boldsymbol{k}} | \alpha \rangle \langle \alpha | \hat{c}_{\sigma,\boldsymbol{k}} | \Psi_0 \rangle \\
&= \sum_{\sigma} \sum_{\alpha} \delta((E_\alpha - E_0) - \omega) \left| \langle \alpha | \hat{c}_{\sigma,\boldsymbol{k}} | \Psi_0 \rangle \right|^2 = \sum_{\alpha} \delta((E_\alpha - E_0) - \omega) s_\alpha(\boldsymbol{k}) \,,
\end{aligned}
\tag{23}
$$

where we defined the spectral weight of each mode as $s_\alpha(\boldsymbol{k})$. For our iPEPS results, we approximate the spectral function by a sum over the spectral weights in the subspace of excitation-ansatz states containing a single B-tensor, with a Lorentzian broadening factor $\eta$. In Eq. 23 we replace the resolution of the identity by a basis of such states $\sum_{\alpha} |\Phi(B_\alpha)_{\boldsymbol{k}}\rangle\langle\Phi(B_\alpha)_{\boldsymbol{k}}|$, where $\alpha$ labels the excitation-ansatz eigenstates. When both the electron and hole parts of the spectrum are taken into account, the following sum rule applies:

$$
\int_{-\infty}^{\infty} \mathrm{d}\omega \left( A_{hl}(\omega, \boldsymbol{k}) + A_{el}(\omega, \boldsymbol{k}) \right) = 2 \,.
\tag{24}
$$

By computing the full spectrum of excitations (i.e. computing all eigenstates in the generalized eigenvalue problem in Eq. 5) we have verified that the sum rule is fulfilled.

The procedure for computing the spectral weights is similar to the calculation of the norm overlap matrix $\mathbb{N}_{\boldsymbol{k}}$ and corresponds to evaluating the following summation:

$$
\begin{aligned}
s_\alpha(\boldsymbol{k}) &= \left| \langle \Phi(B_\alpha)_{\boldsymbol{k}} | \hat{c}_{\sigma,\boldsymbol{k}} | \Psi_0(A) \rangle \right|^2 = \left| \sum_{\boldsymbol{x},\boldsymbol{x}'} e^{i\boldsymbol{k}\cdot(\boldsymbol{x}-\boldsymbol{x}')} \langle \Phi(B_\alpha)_{\boldsymbol{x}} | \hat{c}_{\sigma,\boldsymbol{x}'} | \Psi_0(A) \rangle \right|^2 \\
&= \left| \sum_{\boldsymbol{x},\boldsymbol{x}'} e^{i\boldsymbol{k}\cdot\boldsymbol{x}} \langle \Phi(B_\alpha)_{\boldsymbol{x}} | \Phi(A \cdot c_\sigma)_{\boldsymbol{x}'} \rangle \right|^2 \propto \left| \sum_{\boldsymbol{x}} e^{i\boldsymbol{k}\cdot\boldsymbol{x}} \langle \Phi(B_\alpha)_{\boldsymbol{x}} | \Phi(A \cdot c_\sigma)_{\boldsymbol{0}} \rangle \right|^2 \,,
\end{aligned}
\tag{25}
$$

where $(A \cdot c_\sigma)^j_{\beta,\gamma,\delta,\epsilon} \equiv \sum_i A^i_{\beta,\gamma,\delta,\epsilon} [\hat{c}_\sigma]^{i,j}$ is the result of applying a $\hat{c}_\sigma$ operator on the physical index of a ground-state tensor and $B_\alpha$ is the excitation tensor corresponding to eigenstate $|\alpha\rangle$. Note that the $A \cdot c_\sigma$ tensor lies in the same space and symmetry sector as the excitation $B$ tensors and thus the state $|\Phi(A \cdot c_\sigma)_{\boldsymbol{x}'}\rangle$ falls in the space of excitation-ansatz states. In the last step we again use translational invariance in order to simplify the double summation to a single sum, as in Eq. 19.

Using the summation CTM, the overlap can be calculated. If the full norm matrix has been explicitly computed, the spectral weight can be instantly computed by multiplication with the vectorized representations: $s_\alpha(\boldsymbol{k}) = \left| \vec{\mathbf{B}}^{\dagger}_\alpha \mathbb{N}_{\boldsymbol{k}} \vec{\mathbf{A}}_{c_\sigma} \right|^2$.

In Fig. 3(a) and (c) we show the spectral functions for $U/t = 8$ at bond dimensions $D = 4$ and $D = 5$, respectively, along a path through high-symmetry points of the Brillouin zone,

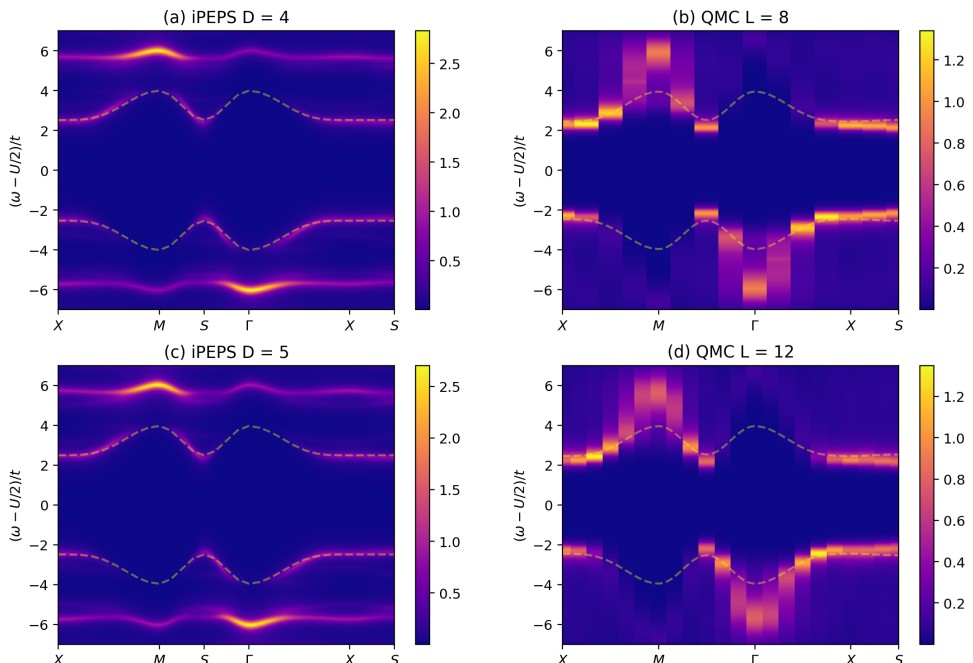

Figure 3: Spectral function at interaction strength $U/t = 8$ along a path $X(\pi,0) - M(\pi,\pi) - S(\pi/2,\pi/2) - \Gamma(0,0) - X(\pi,0) - S(\pi/2,\pi/2)$ through the Brillouin zone for bond dimensions. (a,c) iPEPS results for bond dimensions $D = 4, 5$. (b,d) QMC results for system sizes $L = 8, 12$. The dashed curves in each plot are spin-polaron dispersions [61], fitted to the iPEPS data.

where we apply a Lorentzian broadening factor $\eta = 10^{-2} \cdot (\omega_{\text{MAX}} - \omega_{\text{MIN}})$ scaled by the energy window of the plot. Evidently, the effects of the change in bond dimension are very small, with a change of $\mathcal{O}(10^{-2})$ in energy on the lowest branch, including the gap, and a slightly more clearly defined structure in the spectral function on the higher branches. We note that the particle excitations in the upper half of both figures are related by particle-hole symmetry, with a shift of $\boldsymbol{k} = (\pi,\pi)$. Since the ansatz we use is tailored to single-particle excitations, we limit the energy range of the plot to exclude most higher-lying multiparticle states [6].

The single particle spectral function of a doped hole in a quantum antiferromagnetic is a particularly challenging case study. Spectral weight extends over an energy scale set by the hopping $t$ and at low energy a quasiparticle excitation with dispersion relation set by the magnetic scale $J = 4t^2/U$ emerges [63]. Interpretations of this spectral function range from spin-polaron [61] theories to parton ansätze [55, 57]. Focusing on the hole (negative energy) excitations, we find that the main features of the spectral function agree with results from cluster perturbation theory [64, 65] and quantum Monte Carlo [66, 67]: a low-energy spin-polaron with a sharp dispersion between $-4 < (\omega - {}^U/_2)/t < -2$ and a transfer of spectral weight to a higher-energy branch that peaks around $(\omega - {}^U/_2)/t = -6$ at $\boldsymbol{k} = (\pi,\pi)$. We also plot (dashed lines) the spin-polaron dispersion of the form [61]

$$E^{sp}_{\pm} = \pm\left(x_1 + x_2\left[\cos(k_x) + \cos(k_y)\right]^2\right), \tag{26}$$

with parameters fitted to the iPEPS data ($x_1 = 2.49, x_2 = 0.37$ for $D = 5$, also used in Fig. 3(b,d)).

---

[6]Although the excitation ansatz with a single $B$ tensor is designed for single-particle excitations, some of the effects of multiparticle states can still be approximated up to a certain degree. We note that in order resolve a continuum of excitations more accurately, an excitation ansatz with two independent B tensors could be used, as done in Ref. [62] for MPS.

In Fig. 3 we equally plot the spectral function as obtained by Wick rotating the imaginary time QMC data to the real axis. Here we have used ALF [68] implementation of the stochastic Maximum Entropy algorithm [69], and taken into account the covariance matrix. The QMC data reveals the spin-polaron, and in agreement with the iPEPS data, shows a pronounced loss of spectral weight at the $M(\pi, \pi)$ point at negative energies. In contrast to iPEPS the QMC data captures the continuum nature of the high-energy spectral function. This is particularly clear at the $\Gamma(0, 0)$ point and at frequencies centered around $(\omega - U/2)/t = -6$. As a consequence, resolving the spin-polaron from the continuum of excitations at the $\Gamma(0, 0)$ point is challenging.

## 4 Conclusion

In this paper we have shown how automatic differentiation can be employed in order to greatly simplify the computation of excitations based on the iPEPS excitation ansatz. The algorithm is not only substantially simpler to implement [30], but also more efficient than previous approaches, since it avoids the computationally expensive summation of Hamiltonian terms (which becomes particularly challenging for Hamiltonians with longer-ranged interactions). Instead, the infinite summation is performed for the excitation $B$ tensor and its conjugate, where the derivative with respect to the $B^\dagger$ tensor is computed with AD in an automatized fashion. Furthermore, we have shown that the memory requirements of the algorithm can be significantly reduced by using the fixed-point variant of the CTM summation method. All these improvements make the iPEPS excitation approach more versatile, more efficient, and accessible to a broader community.

We have demonstrated the capabilities of this method by studying the charge gap in the half-filled Hubbard model and comparing our results to quantum Monte Carlo simulations. For $U/t \leq 8$ we found a good agreement between iPEPS and the finite size QMC data, where the iPEPS results approach the extrapolated QMC results with increasing $D$ in a systematic way. At larger $U/t$ obtaining reliable results for the gap with QMC turned out to be exponentially hard, because the scale which needs to be resolved in order to extract the gap is exponentially suppressed with $U/t$. iPEPS does not suffer from this limitation; in fact, the finite $D$ effects become even weaker with increasing $U/t$. We have also presented an example computation of the spectral function $A(k, w)$ for $U/t = 8$ which exhibits only weak finite $D$ effects, and its main features are in agreement with QMC and previous results.

We note that during completion of this work, another AD-based framework to compute excitations with tensor networks has been introduced [70], and tested for MPS in one dimension. This approach, which is based on derivatives of generating functions, is different from the method presented here. Also, very recently there has been progress in applying the MPS excitation ansatz in 2D on cylinders [71].

**Funding information**    This project has received funding from the European Research Council (ERC) under the European Union's Horizon 2020 research and innovation programme (grant agreement No 677061). This work is part of the D-ITP consortium, a program of the Netherlands Organization for Scientific Research (NWO) that is funded by the Dutch Ministry of Education, Culture and Science (OCW). FFA acknowledges financial support from the DFG through the Würzburg-Dresden Cluster of Excellence on Complexity and Topology in Quantum Matter - ct.qmat (EXC 2147, project-id 39085490), through the SFB 1170 ToCoTronics. The quantum Monte Carlo simulations were carried out on the Julia cluster of the University of Würzburg, and the iPEPS simulations on the Dutch national e-infrastructure with the support of the SURF Cooperative.

# A  QMC calculations

Quantum Monte Carlo simulations of the Hubbard model on a square lattice are free of the negative sign problem at the particle-hole symmetric point where, in the notation of Eq. 21, the chemical potential is given by $\mu = U/2$. As a consequence, we will rewrite the Hamiltonian as

$$\hat{H} = -t \sum_{<i,j>,\sigma} \hat{c}^{\dagger}_{i,\sigma}\hat{c}_{j,\sigma} + \text{h.c.} + U\sum_i \big(\hat{n}_{i,\uparrow} - 1/2\big)\big(\hat{n}_{i,\downarrow} - 1/2\big). \tag{27}$$

This rewriting has important consequences since it defines the reference energy from which the energy for addition or removal of an electron is measured. In particular for the above Hamiltonian, the single particle gap is defined as

$$\omega_s = E_0^{N+1} - E_0^N = E_0^{N-1} - E_0^N \tag{28}$$

and takes the value $\omega_s = U/2$ in the atomic, $t = 0$, limit.

We use the projective auxiliary field quantum Monte Carlo method [72–74] to determine the single particle gap. Consider a single slater determinant, $|\Psi_T^N\rangle$, that is required to be non-orthogonal to the ground state, $|\Psi_0^N\rangle$. Here, $N$ is the electron count. The algorithm filters out the ground state from the trial wave function by projection along the imaginary time axis. Hence the ground state expectation value for any observable $\hat{O}$ reads

$$\frac{\langle\Psi_0^N|\hat{O}|\Psi_0^N\rangle}{\langle\Psi_0^N|\Psi_0^N\rangle} = \lim_{\Theta t\to\infty} \frac{\langle\Psi_T^N|e^{-\theta\hat{H}}\hat{O}e^{-\theta\hat{H}}|\Psi_T^N\rangle}{\langle\Psi_T^N|e^{-2\theta\hat{H}}|\Psi_T^N\rangle}. \tag{29}$$

To determine the single particle gap, we consider the observable

$$\hat{O} = \hat{c}_{p,\sigma}(\tau)\hat{c}^{\dagger}_{p,\sigma} \ \text{ with } \ \hat{c}_{p,\sigma}(\tau) = e^{\tau\hat{H}}\hat{c}_{p,\sigma}e^{-\tau\hat{H}} \ \text{ and } \ \hat{c}^{\dagger}_{p,\sigma} = \frac{1}{\sqrt{N}}\sum_i e^{ip\cdot i}\hat{c}^{\dagger}_{i,\sigma}. \tag{30}$$

To at best understand how to extract the single particle gap from the Green function, we consider an energy eigenbasis:

$$\hat{H}|\Psi_n^N(\boldsymbol{p})\rangle = E_n^N(\boldsymbol{p})|\Psi_n^N(\boldsymbol{p})\rangle. \tag{31}$$

Since particle number $\hat{N} = \sum_{i,\sigma}\hat{c}^{\dagger}_{i,\sigma}\hat{c}_{i,\sigma}$ and momentum $\hat{P} = \sum_{p,\sigma}\boldsymbol{p}\,\hat{c}^{\dagger}_{p,\sigma}\hat{c}_{p,\sigma}$ are conserved, the energy eigenvalues can be chosen to satisfy:

$$\hat{P}|\Psi_n^N(\boldsymbol{p})\rangle = \boldsymbol{p}|\Psi_n^N(\boldsymbol{p})\rangle \tag{32}$$

and

$$\hat{N}|\Psi_n^N(\boldsymbol{p})\rangle = N|\Psi_n^N(\boldsymbol{p})\rangle. \tag{33}$$

With the above,

$$G(\boldsymbol{p},\tau) = \langle\Psi_0^N|\hat{c}_{p,\sigma}(\tau)\hat{c}^{\dagger}_{p,\sigma}|\Psi_0^N\rangle = \sum_{n=0}^{\infty} e^{-\tau\big(E_n^{N+1}(\boldsymbol{p})-E_0^N\big)}|\langle\Psi_n^{N+1}(\boldsymbol{p})|\hat{c}^{\dagger}_{p,\sigma}|\Psi_0^N\rangle|^2. \tag{34}$$

Provided that the wave function renormalization, $|\langle\Psi_0^{N+1}(\boldsymbol{p})|\hat{c}^{\dagger}_{p,\sigma}|\Psi_0^N\rangle|^2$ is finite and that $|\Psi_0^{N+1}(\boldsymbol{p})\rangle$ is non-degenerate, then

$$\lim_{\tau\to\infty} G(\boldsymbol{p},\tau) = |\langle\Psi_0^{N+1}(\boldsymbol{p})|\hat{c}^{\dagger}_{p,\sigma}|\Psi_0^N\rangle|^2 e^{-\tau\big(E_0^{N+1}(\boldsymbol{p})-E_0^N\big)}, \tag{35}$$

and we can extract the momentum resolved single particle gap:

$$\omega_s(\boldsymbol{p}) = E_0^{N+1}(\boldsymbol{p}) - E_0^N. \tag{36}$$

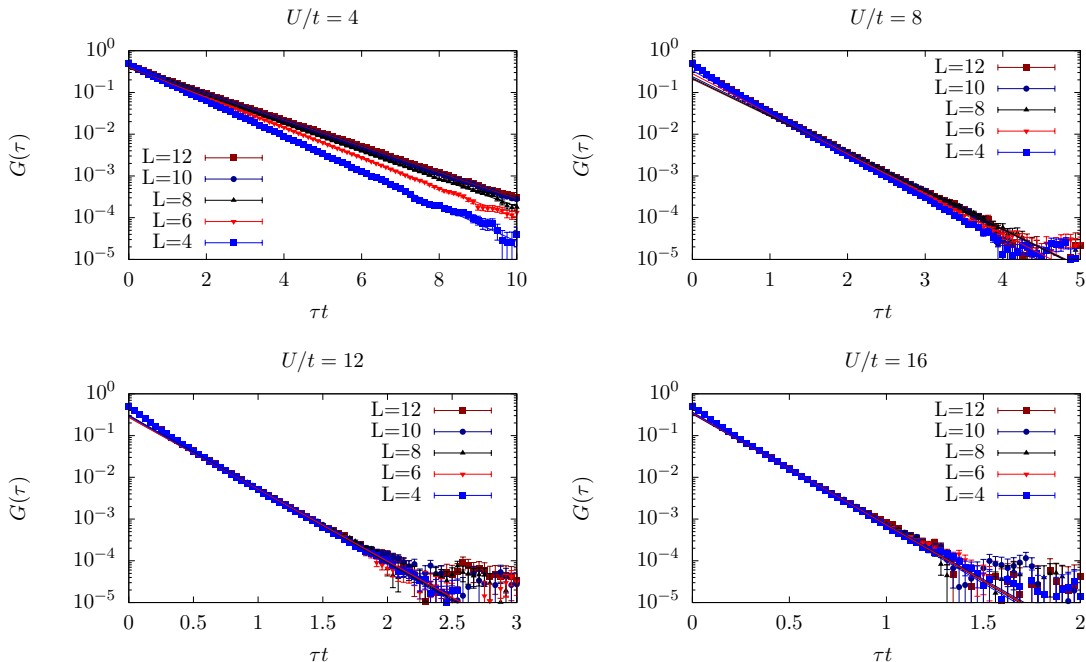

Figure 4: QMC data for the single particle Green function of Eq. 38.

The condition for convergence is

$$e^{-\tau\left(E_1^{N+1}(\boldsymbol{p})-E_0^{N+1}(\boldsymbol{p})\right)} \ll 1. \tag{37}$$

The set of energy differences $E_n^{N+1}(\boldsymbol{p}) - E_0^{N+1}(\boldsymbol{p})$ span the upper Hubbard band, the width of which is set by the energy scale $t$ [63, 75]. Hence we will need $\tau t >> 1$ to ensure a reliable convergence of the results. In the weak to intermediate coupling regime, this is not a problem and for the data set at $U/t = 4$ we could obtain estimates of the gap by setting $\tau_{max}t = 10$. On the other hand, in the strong coupling limit, determining the gap turns out to be an exponentially hard problem. Since we are carrying out simulations at the particle-hole symmetric point, the imaginary time Green function will decay roughly as $e^{-\tau U/2}$ such that reaching say the desired $\tau_{max}t = 10$ will amount to resolving a scale $e^{-10U/t}$. Owing to the central limit theorem this is exponentially expensive in $U/t$. In other words, as $U/t$ grows the QMC results will be increasingly dominated by the *trivial* scale $U/2$ that exponentially damps the relevant information. By construction, this approach will provide an upper bound to the single particle gap.

For our calculations we have considered the quantity:

$$G(\tau) = \frac{1}{N_0} \sum_{\boldsymbol{p},\epsilon(\boldsymbol{p})=0} \langle \Psi_0^N | \hat{c}_{\boldsymbol{p},\sigma}(\tau) \hat{c}_{\boldsymbol{p},\sigma}^\dagger | \Psi_0^N \rangle, \tag{38}$$

with normalization $N_0 = \sum_{\boldsymbol{p},\epsilon(\boldsymbol{p})=0}$ and $\epsilon(\boldsymbol{p}) = -2t\left(\cos(\boldsymbol{p}\cdot\boldsymbol{a}_1) + \cos(\boldsymbol{p}\cdot\boldsymbol{a}_2)\right)$. We have used the ALF-implementation [68] of the projective auxiliary field QMC algorithm to produce the results. Our energy scale is set by $t = 1$ and we have adopted a symmetric Trotter decomposition with $\Delta\tau U = 0.5$. We have used a Hubbard Stratonovich transformation that couples to the density rather than to the z-component of the magnetization. The trial wave function was chosen to be a spin-singlet solution of the non-interacting problem. For this trial wave function a projection parameter $\Theta t = 10$ suffices to reach ground state properties within our error bars. Our data at $U/t = 4, 8, 12$ and $U/t = 16$ is plotted in Fig. 4. As apparent, we

can resolve four orders of magnitude in $G(\tau)$. As $U/t$ grows, the value of $\tau_{max}$ that we can reach within our resolution drops substantially. To estimate the single particle gap, we fit the *tail* of the Green function to a single exponential. For the fit we have taken into account the covariance matrix.

## B Subspace size dependency

In this appendix we discuss the step of restricting the subspace for the $B$ tensors in the excitation ansatz. Generally, the first step would be to restrict the $B$ tensors such that the excited states are orthogonal to the ground state. However, in the simulations for this paper we studied the excitations in a different $U(1)$ symmetry sector than the ground state, therefore this condition is automatically fullfilled for any choice of the parameters in the $B$ tensors.

One important property of the excitation ansatz itself (and is therefore not limited to any particular application), both in one [21] and two [12] dimensions, is the presence of modes with zero norm. Any state with a $B$ tensor of the form

$$B_{\boldsymbol{x}} = e^{i \cdot \boldsymbol{k}} A_{\boldsymbol{x}} \cdot M_{\boldsymbol{x}} - M_{\boldsymbol{x}} \cdot A_{\boldsymbol{x}}, \tag{39}$$

with $M_{\boldsymbol{x}}$ any $D \times D$ matrix, the terms in the momentum superposition will cancel exactly.

This property makes the generalized eigenvalue problem in Eq. 5 ill-defined, so we have to change to a reduced basis in which these null modes have been removed. With this functional

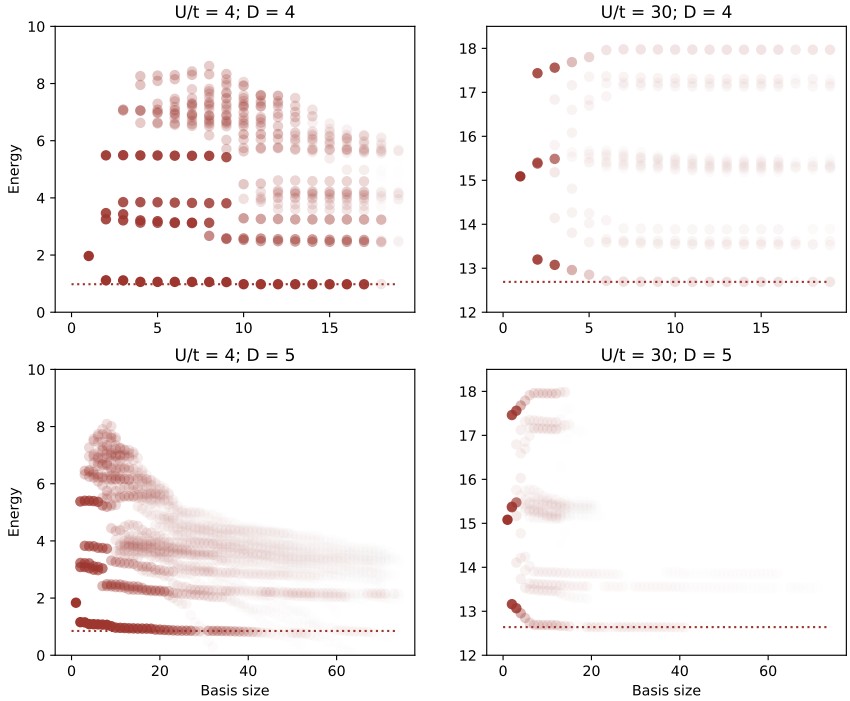

Figure 5: Eigenvalues as a function of basis size, where for each value the generalized eigenvalue problem is solved using a reduced subspace based on the largest eigenvectors of the norm overlap matrix. The opacity of each points is proportional to the corresponding norm.

form for the null modes we can explicitly remove them from the space of the *B* tensors before the simulation starts.

In practice, the removal of the exact null modes does not always guarantee a stable result, and we need to further reduce the subspace by eliminating also modes with a small (but finite) norm. It turns out that, especially in larger-scale simulations, the small-norm modes can lead to spurious eigenvalues when they are present in the subspace. However, since such solutions are characterized by their relatively small norms, generally the correct eigenvalues can be clearly identified.

In Fig. 5 we show the dependence of the eigenvalues on the basis size of the *B*-tensor subspace for several simulations. Note that this information can only be obtained by constructing the full overlap matrices, which is costly compared to an iterative approach. In practice it is enough to check the basis size convergence for only a few selected values at only momenta, depending on the model.

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
