# Peer review of "Automatic differentiation applied to excitations with Projected Entangled Pair States"

_SciPost Physics, doi:SciPost Phys. 12, 006 (2022)_

## Round 1 · Referee Report · Anonymous (Referee 1) · 2021-7-25

Strengths

1) Development of a generic and efficient PEPS based method for computing excitation spectra of strongly correlated systems in 2D 2) Method is clearly presented

Weaknesses

1) Somewhat limited discussion of the spectral functions of the 2D Hubbard model, that are evalu-ated with the proposed technique

Report

In this manuscript, the authors introduce an excitation ansatz for PEPS states that are optimized using automated differentiation (AD) techniques. AD-based techniques have already been successfully used to optimize iPEPS ground states. Here, AD is applied to optimize an excitation ansatz to obtain spectral information of strongly correlated 2D systems; a notoriously challenging task. The proposed method is used to compute the single-particle spectra of the 2D Hubbard model at half filling. The spectra are benchmarked against Quantum Monte Carlo results obtained using Wick rotation techniques (stochastic Max Ent methods).

Overall, the paper is clearly written and the developed computational method is well presented. The discussion of the results is a bit short and could profit from a deeper connection to the existing literature. Here are some suggestion and questions that seem interesting:
1) Could one use rotational symmetry of the state to avoid computing horizontal and vertical optimizations of Eq. 11? In the later example of the Hubbard model a larger unit cell is used, but similar considerations could be transferred to this case too.
2) On page 13 it is briefly discussed why it is hard for QMC to go to large U/t values. As it is written now, it is not immediately clear what is meant without consulting the appendix. It could be beneficial to improve this discussion in the main text.
3) Going from Eq 22 to Eq 23 is exact. It is suggested otherwise in the paragraph in between these two equations. This part should be clarified. Maybe, the convergence generating factor could already be introduced here too.
4) How accurate is the fit to the spin-polaron dispersion in Fig 3? It seems that at the Gamma point, where faint spectral weight is visible in the ground state branch, the spin-polaron dispersion fit strongly overestimates the energy (spin-polaron: w-U/2 ~ -4.5t vs numerics: -3.8t). The authors should discuss implications. Is the proper bandwidth of the two lowest lying branches similar (it looks on the first sight when ignoring the spin-polaron fit).
5) It is surprising that the lowest energy excitation branch is carrying less weight than higher branches. This is somewhat counterintuitive. Do the authors have an intuition why this is so? Is there more spectral weight pushed toward the low energy excitation with increasing D?
6) With increasing U/t, the polaron quasi-particle becomes increasingly extended in real space. What are the implications of using a 2x2 unit-cell for the excitation ansatz?
7) Even though the excitation ansatz is tailored toward extracting low energy properties, often high energy properties are reproduced very nicely too. I would be interested in seeing the spectra at an extended frequency regime (maybe this could be shown in an Appendix?).

Once these questions have been carefully addressed, I think this work is suitable for publication in SciPost, as it develops a novel computational method for computing spectra of 2D strongly correlated systems at zero temperatures.

---

## Round 1 · Referee Report · Johannes Hauschild (Referee 2) · 2021-8-11

Strengths

1) Very clear and (almost) self-contained presentation of the developed method and how it can be implemented.

Report

The authors explain how one can apply automatic differentiation (AD) techniques to optimize 2D tensor networks (PEPS) for the recently introduced excitation ansatz for quasi particles on top of ground state. They benchmark and compare their results for the 2D Hubbard model against unbiased quantum monte carlo (QMC) results and demonstrate that they can comfortably go into the regime $U\gg t$ inaccessible with QMC.

With its detailed explanations, the manuscript enables a broader use of the PEPS-based excitation ansatz by the community. The numerical exploration of spectral properties of strongly-correlated 2D systems is generically very challenging, such that the described technique has the potential to become a state-of-the-art method; further follow-up work based on this technique is to be expected. Hence, the presented paper satisfies the acceptence criteria for Scipost Physics, and I recommend a publication once the following points have been addressed.

Requested changes

1) The explanation of the CTM method is almost self-contained, but it lacks the somewhat important detail of how the projectors to smaller bond dimension around equ. (12) are obtained. Looking into the provided example code reveals that it's basically just an SVD, but I think it would be beneficial to briefly comment on that in the main text as well. Of course, it's the authors decision to which extent they want explain it.

2) I don't understand the pseudo code of Algorithm 2. Line 25 uses apply_H_eff, but only apply_H_eff_fixed_point is defined. On the other hand, the converge_boundaries function is unused. Do you call converge_boundaries at the beginning of apply_H_eff_fixed_point instead of taking C as an extra argument, and use apply_H_eff_fixed_point for the eigensolver?

3) My understanding is that you use an eigensolver that obtains multiple (orthogonal) exciations for each k point at once, correct? Could you maybe stress this in the text? How many excitations did you actually calculate to plot Fig. 3?

4) Upon the first read, it wasn't immeditately clear to me that the $\omega_S$ in Fig. 3 refers to the energy of the excation ansatz at momentum S=(pi/2,pi/2), because the labeled path through the BZ is only introduced later in Fig. 3. Maybe it could be clarified that you know/expect that the excitation energy is minimized at the S point due to the AFM order.

5) I think the term "complete basis of single-particle states" is a bit misleading/confusing/not strictly correct. Really, the basis you use are the excitation ansatz states $|\Phi(B_A)_k>$ on top of the ground state, as becomes evident when you plug them in in eq. 25. These states are still strongly-corrleated many-body wave functions, and they don't form a complete basis - as you also mention in the discussion/comparison with QMC, the PEPS calculations do miss out the higher-energy continuum part of the spectral function, and this is the origin of it.

6) Typos: - ylabel of Fig. 2: comparing with Fig. 3, I think this should also be $(\omega_S - U/2)/t$. - caption of Fig. 3: $S(\pi/2,\pi,2) \rightarrow S(\pi/2,\pi/2)$

---

## Round 1 · Referee Report · Anonymous (Referee 3) · 2021-8-21

Strengths

-Detailed description of the methodology.

Weaknesses

-Just a simple benchmark is reported.
-Some issues on the thermodynamic extrapolations to be fixed.

Report

In this work, the authors develop an efficient method to optimize tensor networks for excited states, with an application to the single-band Hubbard model.

The paper is interesting and well written, with a detailed despription of the optimization part.

I only have a few comments:

1) I do not like (and fully understand) Fig.1. First of all a single inset for two extrapolations (U/t=4 and 12) is a bit confusing, especially because the thermodynamic value for U/t=12 is on the left, while the labels are on the right. In addition, $\omega_s$ is not defined in the main text, but only on the appendix. Most importantly, I suspect that the shift in the main panel is -U/2 and not -U. Otherwise, the values in the inset are not compatible with the ones in the main panel.

2) At page 13, it is written that "the gap can be extracted in a controlled way" for iPEPS simulations. However, in Fig.1 the value of the gap strongly bends down for D=6 and the extrapolated value could be as small as 3.5 or so. I suspect that D=7 or 8 are beyond the present possibilities, but it would be nice to have a fair discussion on the extrapolation issues.

3) Related to the previous points: are the iPEPS gaps reported for the thermodynamic extrapolation or for the largest D values? The thermodynamic extrapolation should have much larger errorbars.

4) I should not understand why in Eq.(23) the upper index "1p" is used. When inserting the complete basis of single-particle states, no approximations are done for the spectral function of Eq.(22).

5) Reference [11] has been published on SciPost Physics in 2021.

In summary, the paper can be published after the authors have addressed these points.

Requested changes

See report.

---

## Round 2 · Referee Report · Anonymous · 2021-10-11

Report

The authors have addressed all my previous points and the paper can be published.

---

## Round 2 · Referee Report · Anonymous · 2021-10-15

Report

The authors have clarified my questions and concerns to a reasonable extend. I recommend to accept the paper for publication.

---

## Round 2 · Referee Report · Johannes Hauschild · 2021-10-18

Report

I'm also satisfied by the authors responses to the questions and recommend a publication.

---

## Round 2 · Author Response

Errors in user-supplied markup (flagged; corrections coming soon)

=============================
Response to the First Referee
=============================

We thank the referee for the consideration of our paper and the comments and suggestions put forward.

1. "Could one use rotational symmetry of the state to avoid computing horizontal and vertical optimizations of Eq. 11? In the later example of the Hubbard model a larger unit cell is used, but similar considerations could be transferred to this case too."

We do not impose rotational symmetry in the excited state, since the $B$ tensor does not possess this symmetry for every value of $k$. Even in the case of rotational symmetric ground state tensors, for k_x ≠ k_y the CTM environment tensors, which contain a sum of B tensors, are not rotational symmetric, and thus imposition of rotational symmetry on the B tensor would constrain the ansatz and lead to a non-optimal solution of the generalized eigenvalue problem.

2. "On page 13 it is briefly discussed why it is hard for QMC to go to large U/t values. As it is written now, it is not immediately clear what is meant without consulting the appendix. It could be beneficial to improve this discussion in the main text."

We have extended the discussion in the main text in order to provide a clearer explanation (for the details we still refer to Appendix A).

3. "Going from Eq 22 to Eq 23 is exact. It is suggested otherwise in the paragraph in between these two equations. This part should be clarified. Maybe, the convergence generating factor could already be introduced here too."

We agree that this part was not clearly formulated. We modified it to separate the exact step of inserting a complete basis of single-particle states and the next step, where we approximate this by a complete basis within the subspace of excitation-ansatz states, and also introduced the Lorentzian broadening factor right after Eq. 23.

4. "How accurate is the fit to the spin-polaron dispersion in Fig 3? It seems that at the Gamma point, where faint spectral weight is visible in the ground state branch, the spin-polaron dispersion fit strongly overestimates the energy (spin-polaron: w-U/2 ~ -4.5t vs numerics: -3.8t). The authors should discuss implications. Is the proper bandwidth of the two lowest lying branches similar (it looks on the first sight when ignoring the spin-polaron fit)."

The fit of the spin-polaron dispersion is based on the iPEPS data, optimized for points with high spectral weight. As a consequence, the points around $\Gamma$ with very small spectral weight deviate more from the fit, while the points around $S$ and $X$ with large spectral weight are more accurately fitted. This leads to a large deviation around the $\Gamma$ point which we agree with the referee is not very satisfactory. Actually, since we do have access to the lowest lying excitation mode and not only the spectral function (in contrast to e.g. QMC), it seems physically more meaningful to perform an unweighted fit, i.e. a fit to the dispersion of the lowest excitation directly. This leads to a much better agreement around the gamma point, while still being reasonably accurate in the regions with large spectral weight. We have modified the plot in the revised version accordingly.

The bandwidth of the low-energy branch is around ~1.5t, while the bandwidth of the higher-energy branch is harder to define. Depending on which threshold of minimal spectral weight to include, we observe a bandwidth between ~1t - 1.5t.

5. "It is surprising that the lowest energy excitation branch is carrying less weight than higher branches. This is somewhat counterintuitive. Do the authors have an intuition why this is so? Is there more spectral weight pushed toward the low energy excitation with increasing D?"

The iPEPS excitation ansatz approximates the spectral function based on a discrete set of single-quasiparticle-like states, and can thus represent a continuum of states only in a limited way. This is the reason why the spectral weight in the higher branch is more localized (resulting in a large spectral weight) instead of being smeared out as in the QMC data. Still, we can observe the tendency that the maximum spectral weight in the higher branch decreases with increasing D, i.e. it gets more smeared out and approximates the continuum more accurately. We note that, in order to resolve a continuum of excitations more accurately, an ansatz with two independent B tensors could be used, which was already tested for matrix product states in [Vanderstraeten et al., PRB 92, 125136 (2015)], but which lies beyond the scope of this work. We have added a comment with this additional reference on p14 to mention this point.

6. "With increasing U/t, the polaron quasi-particle becomes increasingly extended in real space. What are the implications of using a 2x2 unit-cell for the excitation ansatz?"

The choice of unit cell in the ground state size does not relate to any correlation length but only to the partially broken translational symmetry (e.g. a 4x4 unit cell would provide identical results in this case as the 2x2 unit cell). The same is true also for the excitation ansatz. The range of correlations that can be captured is controlled by the bond dimension. We do not observe increasingly stronger finite D effects with increasing U/t, suggesting that the simulations do not become increasingly harder with increasing U/t (unlike in QMC).

7. "Even though the excitation ansatz is tailored toward extracting low energy properties, often high energy properties are reproduced very nicely too. I would be interested in seeing the spectra at an extended frequency regime (maybe this could be shown in an Appendix?)."

In this study we limited ourself to a comparison of the low-energy sector to QMC results, since the iPEPS results are expected to be most accurate there (whereas accurately representing continua of states at higher energies is more challenging). For future work we would be very interested to further explore the higher-energy modes, with a more extended study at higher bond dimensions, and comparison with other methods (such as iPEPS based on a real-time evolution to compute the spectral function).
For reference, we have attached a version of Fig. 3 with a larger energy window for the iPEPS results to this reply.

==============================
Response to the Second Referee
==============================

We thank the referee for the thoughtful questions and suggestions.

1. "The explanation of the CTM method is almost self-contained, but it lacks the somewhat important detail of how the projectors to smaller bond dimension around equ. (12) are obtained. Looking into the provided example code reveals that it's basically just an SVD, but I think it would be beneficial to briefly comment on that in the main text as well. Of course, it's the authors decision to which extent they want explain it."

We note that the computation of the projectors is equivalent for both the ground states and excited states, for which we use the same procedure as in Ref. [29]. We have added an extra sentence to stress this point.

2. "I don't understand the pseudo code of Algorithm 2. Line 25 uses apply_H_eff, but only apply_H_eff_fixed_point is defined. On the other hand, the converge_boundaries function is unused. Do you call converge_boundaries at the beginning of apply_H_eff_fixed_point instead of taking C as an extra argument, and use apply_H_eff_fixed_point for the eigensolver"

These are indeed mistakes, and we thank the referee for pointing them out. Line 25 should use apply_H_eff_fixed_point instead of apply_H_eff. And we should call the converge_boundaries function in order to obtain C before calling the eigensolver.
We have changed the last two lines of the algorithm to fix these points.

3. "My understanding is that you use an eigensolver that obtains multiple (orthogonal) exciations for each k point at once, correct? Could you maybe stress this in the text? How many excitations did you actually calculate to plot Fig. 3?"

This is correct: we formulate the generalized eigenvalue problem separately for each k and solve for the number of required eigenmodes. The total number of possible excitations is bounded by the bond dimension and is equal to the number of parameters in the B tensors (i.e. without any symmetries the number of excitations would be dD^4). For Fig. 3 we calculated the full spectrum in order to check the sum rule. The spectral weight of the two branches is not confined to the lowest two eigenmodes, but there exist several other modes in between. For the energy range of Fig. 3 we need around 15 eigenvalues for each k.

In the revised version we have added a sentence on p.10 to explain more clearly, how higher energy excitations can be obtained, and mention on p.13 that the full spectrum was computed to verify the sum rule.

4. "Upon the first read, it wasn't immeditately clear to me that the $\omega_S$ in Fig. 3 refers to the energy of the excation ansatz at momentum S=(pi/2,pi/2), because the labeled path through the BZ is only introduced later in Fig. 3. Maybe it could be clarified that you know/expect that the excitation energy is minimized at the S point due to the AFM order."

We have added a definition of $\omega_S$ in the caption of Fig. 2, which refers to the gap.

5. "I think the term "complete basis of single-particle states" is a bit misleading/confusing/not strictly correct. Really, the basis you use are the excitation ansatz states $\ket{\Phi(B)_k}$ on top of the ground state, as becomes evident when you plug them in in eq. 25. These states are still strongly-corrleated many-body wave functions, and they don't form a complete basis - as you also mention in the discussion/comparison with QMC, the PEPS calculations do miss out the higher-energy continuum part of the spectral function, and this is the origin of it."

We have changed the text to first show the step of inserting a complete basis of single-particle excitations, which is exact, and afterwards the approximation by a reduced basis in the subspace of excitation-ansatz states.

Typos:
- "ylabel of Fig. 2: comparing with Fig. 3, I think this should also be (ωS−U/2)/t."

The label of Fig. 2 is correct: we shifted by another $(U/2)/t$ to better visualize the differences in the results at the large range of $U/t$ on the x-axis.

- caption of Fig. 3: S(π/2,π,2)→S(π/2,π/2)

We thank the referee for pointing out the typo.

==============================
Response to the Third Referee
==============================

We would like to thank the referee for the comments and questions.

1. "I do not like (and fully understand) Fig.1 [sic]. First of all a single inset for two extrapolations (U/t=4 and 12) is a bit confusing, especially because the thermodynamic value for U/t=12 is on the left, while the labels are on the right. In addition, $\omega_S$ is not defined in the main text, but only on the appendix. Most importantly, I suspect that the shift in the main panel is -U/2 and not -U. Otherwise, the values in the inset are not compatible with the ones in the main panel."

We agree with the referee that the inset could be improved, therefore we split it into two separate figures and used the same shift of $-U/t$ as in the main figure. The shift is actually correct, and we introduced it for better visibility of the energy differences over the large range of U/t values (with a U/2 shift the differences would not be well visible). We have added a comment in Fig.2 to explain our choice of the shift, and also added the definition of $\omega_S$, which was indeed missing.

2. "At page 13, it is written that "the gap can be extracted in a controlled way" for iPEPS simulations. However, in Fig.1 [sic] the value of the gap strongly bends down for D=6 and the extrapolated value could be as small as 3.5 or so. I suspect that D=7 or 8 are beyond the present possibilities, but it would be nice to have a fair discussion on the extrapolation issues."

With “the gap can be extracted in a controlled way” we implied that the gap can be computed for each value of D in a controlled way, and that its accuracy is systematically improving with bond dimension, without facing the issues as with QMC (since it does not rely on a fit to an exponentially small tail as in QMC). To be more clear we replaced “controlled” by “systematic” in the revised version.

We agree with the referee that eventually one would also want to perform a controlled extrapolation of the gap to the infinite D limit. Here we did not attempt to extrapolate based on only 2-3 data points, and because the functional form of how the gap depends on D is not known. Additional values at higher D would indeed be necessary here, however, these large-scale computations lie beyond the scope of this paper, and is thus left for future work (we note that it would also be interesting to explore other ways to extrapolate the gap, e.g. by using a finite correlation length scaling analysis as in PRX 8, 031031, (2018) and PRX 8, 031030, (2018)). We have added a comment on extrapolations in the revised version to mention this point at the end of Sec. 3.1.

3. "Related to the previous points: are the iPEPS gaps reported for the thermodynamic extrapolation or for the largest D values? The thermodynamic extrapolation should have much larger errorbars."

All iPEPS data shown in the figures is computed at different fixed bond dimensions D=4,5,6. The extrapolations and extrapolation error bars in the inset are based on the QMC data. As mentioned above, we did not attempt to perform an extrapolation based on 2-3 data points only.

4. "I should not understand why in Eq.(23) the upper index "1p" is used. When inserting the complete basis of single-particle states, no approximations are done for the spectral function of Eq.(22)."

We have adapted the text in the revised version to first show the exact step of inserting a complete basis of single-particle excitations and then the step of approximating the spectral function with a complete basis in the subspace of excitation-ansatz states, and we have removed the index “1p”.

5. "Reference [11] has been published on SciPost Physics in 2021."

We thank the referee for pointing this out and updated the reference accordingly.

---

## Round 2 · List of Changes

p. 4, Eq (4): removed spurious closing parenthesis in phase factor

p. 8: inserted a sentence
"For the summation CTM we use the same projectors as for the regular CTM, for which we follow the procedure of Ref.[29], with the additional remark that the QR decomposition used in Ref.[29] is not necessary for the computations [40]."
with "[40] T. Okubo, private communication" added to the bibliography

p. 9, Eq (19): changed “.” -> “,”

p. 10:
“Once the actions of the effective matrices are implemented, an iterative solver for generalized eigenvalue problems can be used to compute the lowest excited states, described in~\cref{alg:full_ev}.”
->
“Once the actions of the effective matrices are implemented, an iterative solver for generalized eigenvalue problems can be used to compute the lowest excited state for a fixed value of k, described in~\cref{alg:full_ev}.
Higher-energy excitations can be obtained by solving the generalized eigenvalue problem for the $k$ lowest eigenvalues $w_\alpha$, resulting in different tensors $B_\alpha$, where $\alpha=\{1,2,\ldots,k\}$ labels the eigenstates.”

p. 11, Algorithm 2: inserted new line "$\mathcal{C} \gets$ converge_boundaries"

p. 11, Algorithm 2, line 26:
"apply\_$H_{eff}"
->
"apply\_$H_{eff}$\_fixed\_point"

p. 13, Fig. 2: split the inset into two separate figures to improve the readability, and inserted a sentence in the caption:
"On the y axis, $\omega_S$ refers to the lowest eigenvalue at the $S=(\pi/2,\pi/2)$ point, and we applied a shift of $-U/t$ to better visualize the energy differences over the large range of $U/t$ on the x axis."

p. 13:
"The reason is that the relevant scale which needs to be resolved to extract the gap is exponentially suppressed with $U/t$, see \cref{sec:qmc_calculations} for details."
->
"The reason is that the tail of the single particle Green function $G(\tau)$, from which the gap is extracted at large $\tau$, is exponentially suppressed with $U$, such that the number of Monte Carlo samples needed to resolve the tail grows exponentially with $U/t$, see Appendix A for details."

p. 13:
“the gap can be extracted in a controlled way”
->
“the gap can be extracted in a systematic way”

p. 14: inserted
“We note that, while the data systematically improves with increasing $D$, we have not attempted to extrapolate the gap to the infinite $D$ limit based on only 2-3 data points, also because the functional form of how the gap depends on $D$ is not known a priori.”

p. 14:
"Since our method is mainly applicable to single-particle (e.g. single electron or hole) states, we compute a single-particle approximation of the full spectral function by inserting a complete basis of single-particle states, $\sum_{\alpha} \dyad{\alpha}$ with momentum $\vb*{k}$:
<eq 23>
where we defined the spectral weight of each mode as $s_\alpha(\vb*{k})$."
->
"By inserting a complete basis of single-particle states $\sum_{\alpha} \dyad{\alpha}$ with momentum $\vb*{k}$ we obtain
<eq 23>
where we defined the spectral weight of each mode as $s_\alpha(\vb*{k})$.
For our iPEPS results, we approximate the spectral function by a sum over the spectral weights in the subspace of excitation-ansatz states containing a single B-tensor, with a Lorentzian broadening factor $\eta$.
In Eq. 23 we replace the resolution of the identity by a basis of such states $\sum_{\alpha} \dyad{\Phi(B_{\alpha})_{\vb*{k}}}$, where $\alpha$ labels the excitation-ansatz eigenstates."

p. 14: inserted
“By computing the full spectrum of excitations (i.e. computing all eigenstates in the generalized eigenvalue problem in~\cref{eq:eig_prob}) we have verified that the sum rule is fulfilled.”

p. 14, Eqs 23 and 24: removed “1p” superscript from the spectral function

p. 15, Fig. 3: changed the fit to be unweighted by the spectral weight, as well as reducing the fitting parameters to x1 and x2

p.15, caption of Fig. 3:
"S(\pi/2,\pi,2)" -> "S(\pi/2,\pi/2)"

p.15, footnote:
“Although the excitation ansatz with a single $B$ tensor is designed for single-particle excitations, some of the effects of multiparticle states can still be approximated.”
->
“Although the excitation ansatz with a single $B$ tensor is designed for single-particle excitations, some of the effects of multiparticle states can still be approximated up to a certain degree. We note that in order resolve a continuum of excitations more accurately, an excitation ansatz with two independent B tensors could be used, as done in Ref. [L. Vanderstraeten, F. Verstraete, and J. Haegeman, Scattering Particles in Quantum Spin Chains, Phys. Rev. B 92, 125136 (2015).] for MPS.”

p. 16:
removed the x3 term from Eq (26) and changed
“with parameters fitted to the iPEPS data ($x_1=2.43,x_2=0.50,x_3=0.012$ for $D=5$, also used in~\cref{fig:full_E_spec}(b,d)).”
->
“with parameters fitted to the iPEPS data ($x_1=2.49,x_2=0.37$ for $D=5$, also used in~\cref{fig:full_E_spec}(b,d)).”

p. 16:
“a the $\Gamma(0,0)$ point”
->
“at the $\Gamma(0,0)$ point”

Ref. [11]: updated arXiv version to SciPost publication citation

---

## Editorial Decision

published